# Osteogenic Properties of Novel Methylsulfonylmethane-Coated Hydroxyapatite Scaffold

**DOI:** 10.3390/ijms21228501

**Published:** 2020-11-12

**Authors:** Jeong-Hyun Ryu, Tae-Yun Kang, Hyunjung Shin, Kwang-Mahn Kim, Min-Ho Hong, Jae-Sung Kwon

**Affiliations:** 1Department and Research Institute of Dental Biomaterials and Bioengineering, Yonsei University College of Dentistry, Seoul 03722, Korea; sweetden623@gmail.com (J.-H.R.); tykang@yuhs.ac (T.-Y.K.); kmkim@yuhs.ac (K.-M.K.); 2BK21 PLUS Project, Yonsei University College of Dentistry, Seoul 03722, Korea; 3Nature Inspired Materials Processing Research Center, Department of Energy Science, Sungkyunkwan University, Suwon 16419, Korea; hshin@skku.edu

**Keywords:** methylsulfonylmethane, hydroxyapatite, scaffold, osteoinduction

## Abstract

Despite numerous advantages of using porous hydroxyapatite (HAp) scaffolds in bone regeneration, the material is limited in terms of osteoinduction. In this study, the porous scaffold made from nanosized HAp was coated with different concentrations of osteoinductive aqueous methylsulfonylmethane (MSM) solution (2.5, 5, 10, and 20%) and the corresponding MH scaffolds were referred to as MH2.5, MH5, MH10, and MH20, respectively. The results showed that all MH scaffolds resulted in burst release of MSM for up to 7 d. Cellular experiments were conducted using MC3T3-E1 preosteoblast cells, which showed no significant difference between the MH2.5 scaffold and the control with respect to the rate of cell proliferation (*p* > 0.05). There was no significant difference between each group at day 4 for alkaline phosphatase (ALP) activity, though the MH2.5 group showed higher level of activity than other groups at day 10. Calcium deposition, using alizarin red staining, showed that cell mineralization was significantly higher in the MH2.5 scaffold than that in the HAp scaffold (*p* < 0.0001). This study indicated that the MH2.5 scaffold has potential for both osteoinduction and osteoconduction in bone regeneration.

## 1. Introduction

The regeneration of tissue in large bone defects requiring oral, craniofacial, or orthopedic surgery is limited and remains a substantial therapeutic challenge [1]. The use of autologous bone grafts for bone regeneration has been the gold standard, given its strong osteogenic potential [2]. However, the supply of autologous bones to patients with damaged bones is limited [3]. Thus, the development of bone substitute materials, such as hydroxyapatite (HAp), β-tricalcium phosphate, and biphasic calcium phosphate, for bone regeneration has been proven to be efficient [4].

Studies have shown that HAp is the main inorganic mineral component of bone and teeth [5,6]. Hydroxyapatite has good biocompatibility and osteoconduction ability as an artificial bone substitute material [7,8]. In addition, the material allows adhesion of bone cells and cell proliferation [9]. However, despite its excellent osteoconductive properties, HAp has limited application owing to weak osteoinduction [10,11].

Methylsulfonylmethane (MSM, C_2_H_6_O_2_S), also known as dimethyl sulfone, is an organic sulfur molecule. MSM is a naturally derived sulfur compound found in many plants and food sources [12,13]. It is non-toxic, has anti-inflammatory activity, and has been shown to prevent osteoarthritis. Recently, it was shown that MSM could promote osteogenic differentiation of mesenchymal stem cells (MSCs) [14,15]. Kim et al. showed that MSM increases the expression of bone morphogenetic protein 2 (BMP-2) to induce the differentiation of MSCs into osteoblasts [16]. This indicated that MSM might have the ability to induce osteogenic differentiation. Although MSM was well known in osteogenic differentiation, there has only been a few reports for the application of MSM in the biomedical field, such as bone tissue engineering and regenerative medicine.

Dip-coating is the simplest technique that does not require substantial machinery and equipment [17]. In this method, the porous scaffold is dipped into the coating solution containing a specific drug-like chemical that would allow the ceramic scaffold to act as a carrier for the drug delivery system [18,19,20]. Many studies have demonstrated that scaffolds coated with the drug solution using the dip-coating technique are effective for bone tissue engineering [21,22,23].

Therefore, in this study, an MSM aqueous solution coated on the HAp scaffold (MH) was successfully fabricated by the sponge replica and dip-coating method. This study was based on the hypothesis that the MH scaffold will induce considerable osteoblast proliferation and differentiation with respect to the control HAp scaffold.

## 2. Materials and Methods

### 2.1. Materials

Nano sized HAp powder was purchased from Ossgen (Daegu, Korea). Polyvinyl butyral (PVB) and MSM purchased from Sigma-Aldrich (St. Louis, MO, USA) were used. Polyvinylidene difluoride membranes obtained from Millipore (Schwalbach, Germany) were used. 3-(4,5-Dimethylthiazol-2yl)-2,5-diphenyltetrazolum bromide (MTT) and alizarin red S purchased from Sigma-Aldrich (St. Louis, MO, USA) were used. All chemicals and reagents were used as received without further purification unless otherwise noted.

### 2.2. HAp Scaffold Preparation

The hydroxyapatite scaffold was prepared in accordance with the method described previously [11]. In brief, 5% (*w/v*) PVB in a solution (binder) of absolute ethanol was vigorously stirred for 2 h and added to 55% (*w/v*) HAp solution. The mixture was vigorously stirred for an additional 24 h. Polyurethane (PU) foam templates were punched to form a three-dimensional cylindrical shape (diameter, 7 mm; height, 9 mm) and immersed in the HAp slurry. In order to make the porous scaffold, the slurry was dispersed constantly and the blocked pores were removed using an air gun. The sponge-coated HAp slurry was then dried at 90 °C for 30 min. These dipping and drying steps were repeated twice. The sponge-coated HAp slurry was heated to 600 °C at a heating velocity of 5 °C/min for 1 h in order to burn out the sponge and binder. Finally, the sample was sintered to 1250 °C for 3 h at a heating velocity of 5 °C/min using a furnace (Lindberg/MPH, Riverside, MI, USA).

### 2.3. MH Scaffolds Preparation

Hydroxyapatite scaffolds were coated with MSM by the dip-coating/vacuum storage method, in accordance with the method described previously [24,25]. MSM-coated samples were stored in a vacuum chamber for 24 h, followed by drying at 60 °C overnight in a dry oven. The drying temperature was fixed based on the melting point of MSM, which has been previously described [26]. The concentration of methylsulfonylmethane used in the dip-coating process differed depending on the experimental groups, which included 2.5, 5, 10, and 20% (*w/v*) solutions prepared by dissolving MSM into distilled water (Table 1).

### 2.4. Scanning Electron Microscopy Observation

The morphological characteristics of the scaffolds were observed by scanning electron microscopy (SEM; MERLIN, Carl Zeiss AG, Oberkochen, Germany) at an accelerating voltage of 15 kV, after sputter coating with carbon. In addition, an elemental mapping analysis was performed by energy dispersive spectroscopy (XFlash 6130, Bruker, Berlin, Germany) on the elemental composition of the scaffolds.

### 2.5. In Vitro Sulphur Ion Release

The in vitro release of MSM from MH scaffolds was considered by the method adapted from previously published literature [27]. Both HAp and MH scaffolds were immersed in 10 mL of phosphate-buffered saline (PBS) and incubated for 14 d. PBS was then removed and filtered (pore size, 0.2 μm). Sulfur ion levels were determined using an inductively coupled plasma–optical emission spectrophotometer (ICP–OES; iCAP7400, Thermo Fisher Scientific, Waltham, MA, USA), and the ions were considered to be an indicator for MSM.

### 2.6. Cell Subculture

MC3T3-E1 osteoblastic cells (CRL-2593, subclone 4) from mouse calvaria obtained from the American Type Culture Collection (Manassas, VA, USA) were used, which were cultured in alpha minimum essential medium (α-MEM; Welgene, Gyeongsan, Korea). Each culture medium consisted of 10% fetal bovine serum (FBS; Gibco, Grand Island, NY, USA) and 1% antibiotic/antimycotic mix and the completed culture medium was changed every 2 to 3 d. Furthermore, the osteogenic medium was induced by the addition of 50 μg/mL ascorbic acid, 10 mM β-glycerophosphate, and 100 nM dexamethasone. The osteogenic medium was subsequently replaced every 2 d.

### 2.7. Cell Proliferation

MC3T3-E1 cells were seeded at a density of 1 × 10^5^ cells per scaffold (in a 24-well culture plate) and cultured for 3 days. Cell proliferation was quantitatively analyzed using the MTT (1 mg/mL) assay in accordance with the method described in a previous study [28]. Briefly, cells cultured on the 24-well culture plate were exposed to test and control materials indirectly as the materials were placed in a cell insert, so that extractants from the materials would be in contact with the cells. The optical density (O.D) was measured at 570 nm using a microplate spectrophotometer (Epoch, Bio-Tek, Seoul, Korea). The cell proliferation was calculated as follows:Cell proliferation rate (%) = [O.D] _day 3_ − [O.D] _day 1_/[O.D] _day 1_ × 100(1)

### 2.8. Alkaline Phosphatase (ALP) Activity

To evaluate ALP activity, the indirect method described in the cell proliferation method was applied in accordance with a previous study [28]. The test was performed for 4 and 10 days to demonstrate the initial step of osteogenic differentiation. ALP activity was measured by using a SensoLyte para-Nitrophenylphosphate (pNPP) alkaline phosphatase assay kit (Anaspec, San Jose, CA, USA) in accordance with the manufacturer’s protocol. The total protein content was measured by using a Pierce bicinchoninic acid assay kit (Thermo Fischer Scientific, Waltham, MA, USA) following the manufacturer’s protocol.

### 2.9. Mineralization

To evaluate mineralization, the indirect method described in the cell proliferation method was applied in accordance with a previous study [28]. The deposition of calcium at day 14 from the osteoblast-like cells was considered by alizarin red staining, in accordance with the method described earlier [11]. In brief, cells were cultured, attached to the cell culture plate, rinsed with PBS, and fixed in 70% EtOH at 4 °C for 1 h. The fixed cells were then stained with 40 mM alizarin red (pH 4.2) by incubating with the solution for 10 min at room temperature. Plates were washed with deionized water until the point where the dye’s colour disappears. The residual stain with 10% cetylpyridinium chloride was eluted and the optical density of this solution was measured at 562 nm using a microplate spectrophotometer (Epoch, Bio-Tek, Seoul, Korea).

### 2.10. Statistical Analysis

The graphical data are presented as mean ± standard deviation. Experimental data of the in vitro test were analyzed statistically by one-way analysis of variance with Tukey’s post–hoc tests (SPSS23, Chicago, IL, USA). The statistical significance was set at a *p* value < 0.05.

## 3. Results

### 3.1. Characterization of HAp and MH Scaffold

The surface morphology and structure of HAp and MH groups observed using scanning electron microscopy (SEM) are shown in Figure 1. The macroporous structures of HAp scaffolds showed a degree of interconnectivity, which is a characteristic feature of porous scaffolds formed by the sponge replica method [29]. The SEM images in the HAp group showed a preserved microporous structure and indicated the presence of Ca and P. However, MH groups indicated Ca, P, and S. We confirmed that the MH groups were successfully coated with the aqueous MSM solution in the HAp group.

### 3.2. Sulphur Ion Release Test

The release curves of MH scaffolds in phosphate-buffered saline (PBS) are shown in Figure 2. There was an initial burst release of sulfur ions in MSM for MH10 and MH20 groups at day 1, which was significantly greater than that observed in case of MH2.5 and MH5 groups (*p* < 0.0001). All the groups showed constant release of MSM until day 7. No sulfur ions in MSM were detected on day 21.

### 3.3. Cell Proliferation in HAp and MH Scaffolds

Cell proliferation and calculated proliferation rate in the HAp and MH groups from day 1 to day 3 are shown in Figure 3. The cell proliferation rate in the HAp group was not significantly different from that in the MH2.5 group (*p* > 0.05). Meanwhile, there was a significant increase in the cell proliferation rate in the MH2.5 group compared with that observed in MH5 and MH10 groups (*p* < 0.05).

### 3.4. ALP Activity in HAp and MH Scaffolds

The ALP activity of MC3T3-E1 cells from HAp and MH groups on day 4 and day 10 was assessed. The ALP activity of HAp and MH groups was not significantly different at day 4. However, the ALP activity of MH2.5 group was significantly higher than that of HAp and MH5 groups at day 10 (*p* < 0.0001), while the ALP activity of MH5 was significantly higher than that of HAp (Figure 4).

### 3.5. Mineralization in HAp and MH Scaffolds

The effect of released ions from MH groups on the mineralization of MC3T3-E1 cells over 14 d was investigated (Figure 5). Significantly higher levels of calcium deposition (*p* < 0.05) were observed in the MH2.5 group than in the other groups. However, mineralization in the MH5 group was significantly lower than that in HAp and MH2.5 groups (*p* < 0.0001).

## 4. Discussion

Here, we successfully coated the HAp scaffold with MSM using the dip-coating method, which demonstrated an adequate level of biocompatibility and osteoblastic proliferation and differentiation linked to potential hard tissue regeneration.

The MSM coating on HAp scaffolds maintained its porous bone-like structure (Figure 1) [11]. Moreover, we confirmed the presence of sulfur ions from MSM on the surface of MH scaffolds. The dip-coating method is recognized as an effective method for chemical coating of HAp scaffolds [30,31]. Hence, in this study, dip-coating was used to coat the scaffolds.

The osteoinductive effect is dependent on MSM present on the surface, and that which is released from the HAp scaffold. In Figure 2, we demonstrated that MSM release was detected in MH scaffolds. MSM was sharply released from the MH10 and MH20 groups on day 1. Burst release can be possibly attributed to the exposure of MSM present on the outer surface of the HAp scaffold. This would explain the initial release [27]. In addition, MSM was continuously released from MH scaffolds until day 7.

In Figure 2, we demonstrated that such a burst release of MSM could be cytotoxic where high doses of MSM were used (group MH20). Kim et al. reported that exposure of >10 mg/mL MSM to cells for 24 h was toxic [32]. Further, Ezaki et al. reported that a high dose intake of MSM (0.6 and 6 g/kg/day in rats) caused adverse effects, such as reduction of body and tissue weight [12]. Clinical trials have indicated that MSM intake causes nausea, diarrhea, headache, and eye irritation in some patients [33,34].

When a high level of MSM was released from MH scaffolds, it was evident that the cell proliferation rate would also be influenced by a certain dose of MSM (Figure 3). In groups MH5 and MH10, the level of MSM released from the MH scaffolds resulted in the inhibition of cell growth in comparison to that observed in the HAp group. These observations were in agreement with a previous report by Wang et al. in which lower bioactivity and disturbed cell proliferation were observed with 10 wt.% MSM in electrospun poly (lactic-co-glycolic acid) fibers compared with controls [27]. A possible reason for this decrease in cell proliferation rate could be the level of sulfur present in MSM. Several studies have reported that the inhibition of cell growth was due to a high level of sulfur in MSM [35,36]. Based on the results of sulfur ion release test and cell proliferation assays, we performed cell mineralization studies with the remaining groups, excluding MH10 and MH20.

Despite cellular cytotoxicity to cells and a disturbed cellular proliferation rate at a certain level of MSM (MH2.5), that osteoinductive features of MSM were evident, as indicated by ALP activity (Figure 4) and mineralization of the preosteoblasts (Figure 5). The ALP activity of the MH2.5 group was higher than that of HAp and MH5 groups at day 10. In this result, the MH2.5 group induced the differentiation from preosteoblasts to osteoblasts. Kim et al. reported that MSM induced the expression of bone morphogenetic proteins for osteoblast differentiation in MSCs [16]. While MSM had a positive effect on osteogenic differentiation from MSCs to osteoblasts, a previous study had confirmed that MSM increased the activity of ALP and the expression of osteonectin, osteocalcin, osterix, and Runx2. Furthermore, MSM enhanced the growth hormone, which promoted Janus kinase/signal transducers and activators of transcription pathway in osteoblast differentiation [14].

Mineralization is the final stage of osteogenesis. Based on the ALP activity result, most mineralization was observed in the MH2.5 group at day 14. According to mineralization results in this study, MSM enhanced the mineralization effect from stem cells to osteoblast-like cells, as shown in stem cells from human exfoliated deciduous teeth [15]. In terms of mineralization, the main calcium component was well known to easily combine with alizarin red [37]. There is a limitation to using calcium phosphate-based materials directly in cells. However, it was suggested as an indirect method to demonstrate mineralization in this study.

Although MC3T3-E1 preosteoblasts can differentiate into osteoblasts [38], there are limitations with respect to proving their osteogenic effects using in vitro assays as a single cell line cannot completely recapitulate the clinical environment [39].

This study was a limited in vitro study that did not consider the clinical outcomes of using the MSM-coated HAp. Properties such as mechanical and physical features were not considered in this study, which could be a part of future studies. However, the study clearly showed possibilities of using the MSM-coated HAp, especially MH2.5 scaffolds, for improved osteoinduction in hard tissue regeneration.

## 5. Conclusions

We successfully fabricated a HAp scaffold with MSM using the dip-coating method and facilitated the use of MSM in the biomedical field. The obtained MH scaffold did not affect the porosity of the HAp scaffold. Controlled and continuous release of MSM in the MH scaffold was observed until day 7 and the bioactivity of MSM was retained. We confirmed that the MH2.5 scaffold was biocompatible, using cytotoxicity and cell proliferation assays, and that it was effective at increasing the osteogenic potential, as demonstrated by calcium deposition of MC3T3-E1 cells. Taken together, the MH2.5 scaffold can serve as a bone substitute to support the restoration of bone structure and damaged bone.

## Figures and Tables

**Figure 1 ijms-21-08501-f001:**
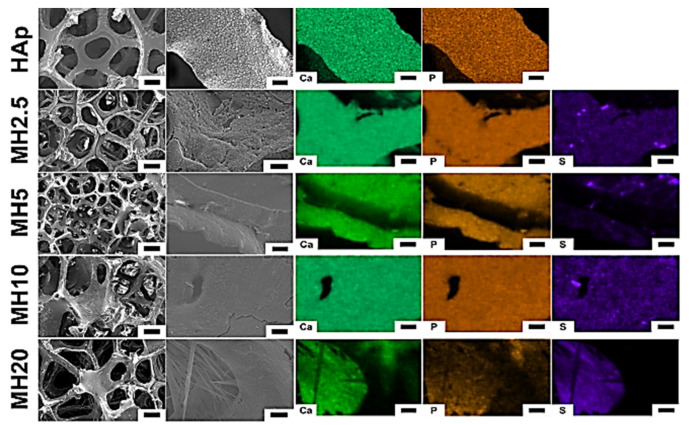
Surface morphology (left) and chemical elements (right) of porous hydroxyapatite (HAp) and MH groups. Element mapping analyses were performed for calcium, phosphorous, and sulfur. Low magnification (100×) images: scale bar is 200 μm, high magnification (1000×) images: scale bar is 10 μm.

**Figure 2 ijms-21-08501-f002:**
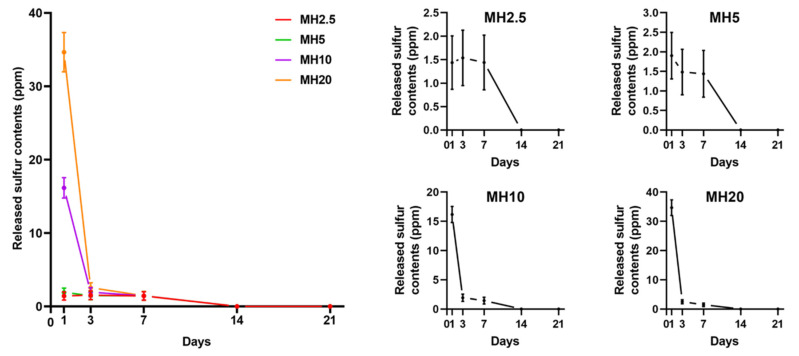
Sulfur ions in methylsulfonylmethane released from different MH groups into phosphate-buffered saline (PBS), in accordance with the day(s) of immersion (*n* = 9).

**Figure 3 ijms-21-08501-f003:**
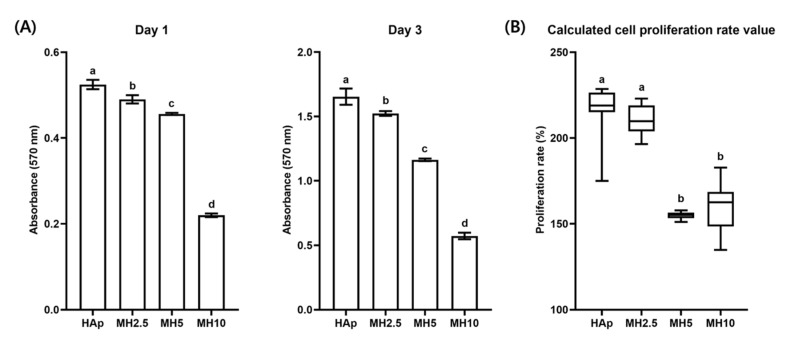
(**A**) Cell proliferation and (**B**) calculated cell proliferation rate in porous hydroxyapatite (HAp) and MH scaffolds. Bars labeled with different letters indicate significant differences (*n* = 8, *p* < 0.05).

**Figure 4 ijms-21-08501-f004:**
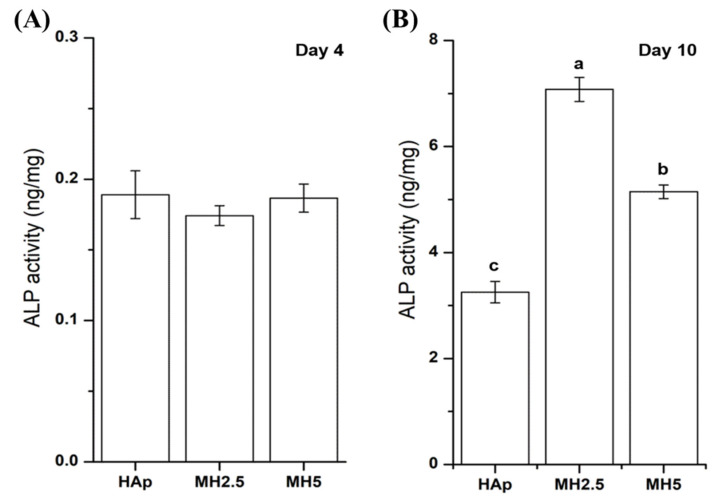
Alkaline phosphatase activity of MC3T3-E1 cells in porous hydroxyapatite (HAp) and MH groups after 4 days (**A**) and 10 days (**B**) for initial osteogenic differentiation. Bars labeled with different letters indicate significant differences (*n* = 6, *p* < 0.05).

**Figure 5 ijms-21-08501-f005:**
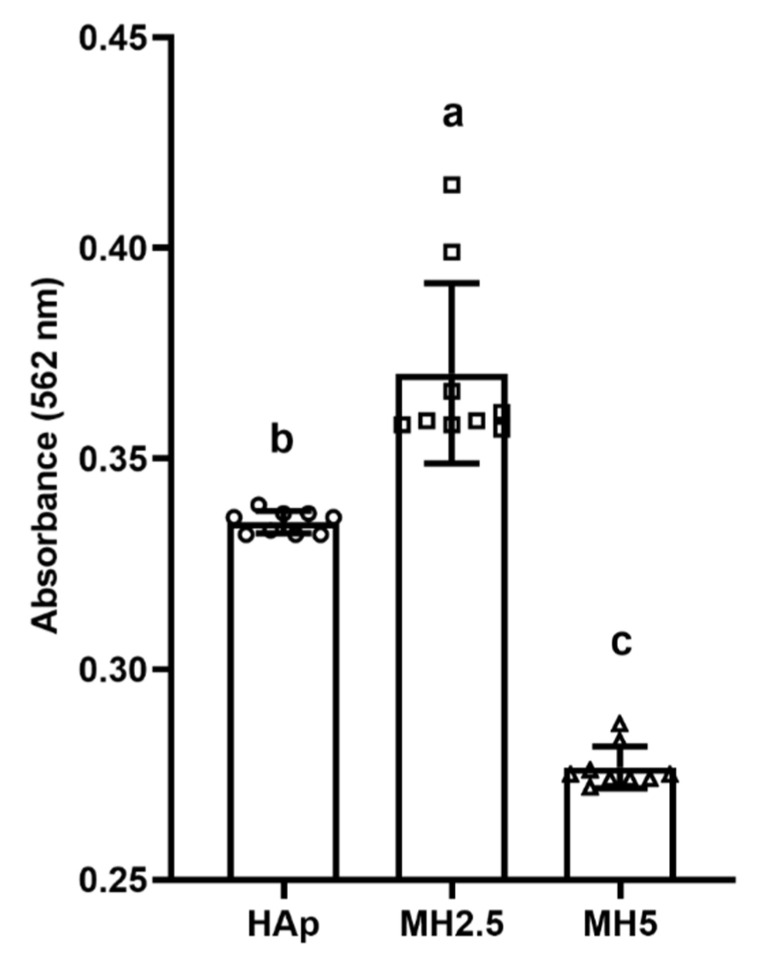
Quantitative analysis of calcium deposition in porous hydroxyapatite (HAp) and MH groups based on alizarin red staining for osteogenic differentiation. Bars labeled with different letters indicate significant differences (*n* = 9, *p* < 0.05).

**Table 1 ijms-21-08501-t001:** Methylsulfonylmethane concentrations used for the control and experimental groups.

MSM Concentration (*w/v*)	0%	2.5%	5.0%	10.0%	20.0%
Group Code	HAp	MH2.5	MH5	MH10	MH20

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
