# Peer review of "Osteogenic Properties of Novel Methylsulfonylmethane-Coated Hydroxyapatite Scaffold"

_ijms, 2020, doi:10.3390/ijms21228501_

Round 1

Reviewer 1 Report

The Manuscript entitled "Osteogenic properties of novel  methylsulfonylmethane-coated hydroxyapatite scaffold" described that MH2.5 scaffold has potential of both osteoinduction and osteoconduction in bone regeneration. This manuscript is well written and nicely presented. I would recommend minor revision for the same. Comments are mentioned below.

  1. There are many other chemicals that contain sulfur ions, so why did you use methylsulfonymethane? Does methylsulfnymethane have a special chemical mechanism in the osteoinduction process than other chemicals containing sulfur ions? I recommend you to add this to the discussion section.
  2. The letters and scale bars shown in Figure 1 are not displayed correctly. Express it in Figure Legend or make it clearly visible in the Figure. Also, author referred to use the EDS in 2.4 Scanning electron microscopy observation. However, the author showed the element mapping in the Figure 1. Please revise by changing the sentence from EDS to element mapping in 2.4 Scanning electron microscopy observation and Figure 1.
  3. In Figure 6, it is not stated what (A) and (B) are. You should add the scale bar in Figure 6(A). Scale bar and text in the image is not looking clear.
  4. Some of the references are out of style.

Author Response

Review #1

The Manuscript entitled "Osteogenic properties of novel methylsulfonylmethane-coated hydroxyapatite scaffold" described that MH2.5 scaffold has potential of both osteoinduction and osteoconduction in bone regeneration. This manuscript is well written and nicely presented. I would recommend minor revision for the same. Comments are mentioned below

Comment #1

There are many other chemicals that contain sulfur ions, so why did you use methylsulfonymethane? Does methylsulfnymethane have a special chemical mechanism in the osteoinduction process than other chemicals containing sulfur ions? I recommend you to add this to the discussion section

Response to Comment #1

Thank you for your valuable comments. MSM is well known natural compound that is biocompatibile for the use in human body [Butwan et al]. Additionally, previous study by Joung et al demonstrated that methylsulfonylmetane enhanced the growth hormone and it promoted the Janus kinase/signal transducers and activators of transcription pathway in the osteoblast differentiation [14]. Such ideal biological features as well as possibilities of coating on hydroxyapatite led to the idea of this study. We now have added these details to the discussion section line 265 to 268.

Comment #2

The letters and scale bars shown in Figure 1 are not displayed correctly. Express it in Figure Legend or make it clearly visible in the Figure. Also, author referred to use the EDS in 2.4 Scanning electron microscopy observation. However, the author showed the element mapping in the Figure 1. Please revise by changing the sentence from EDS to element mapping in 2.4 Scanning electron microscopy observation and Figure 1.

Response to Comment #2

Thank you for kind your comment. We agree that the letter and scale bars are not correct in the Figure 1, and sorry for such mistakes. We now have revised the EDS to elemental mapping in the Figure 1.

Comment #3

In Figure 6, it is not stated what (A) and (B) are. You should add the scale bar in Figure 6(A). Scale bar and text in the image is not looking clear.

Response to Comment #3

Thank you for kind your comment. We now included details on (A) and (B) in the Figure 6. Also, in accordance with your comment, we revised the scale bar and text in the image to make them visibly clear.

Comment #4

Some of the references are out of style.

Response to Comment #4

Thank you for kind your comment. As followed by your comment, we revised the references to suit the style

Reviewer 2 Report

Basically this article is well organized. Only some minor details need to be added:

  1. what are the solvents for PVB / HA and dip-coating for MSM
  2. You need to mention why only MH2.5 is tested in animal trial and also no MH10 and MH20 in 3.4 and 3.5.
  3. You have tested sulphur concentration by using ICP-OES. So, does it mean the release of sulphur ion or MSM? 
  4. Fig 1 - different colours represent which colours?

Author Response

Review #2

Basically, this article is well organized. Only some minor details need to be added:

Comment #1

What are the solvents for PVB / HA and dip-coating for MSM

Response to Comment #1

Thank you for kind your comment. In this study, we used the absolute ethanol in the solvent component of PVB solution according to reference.[11] On the other hand, we used the distilled water in the solvent component of MSM aqueous solution as with previous literatures stating that MSM is suitable for polar solvent such as water, acetone, alcohol. We now have included this in Materials and Methods.

Comment #2

You need to mention why only MH2.5 is tested in animal trial and also no MH10 and MH20 in 3.4 and 3.5.

Response to Comment #2

Thank you for kind and encouraging comment. We have used MH2.5 as both ALP activity and mineralization study showed it to be the optimal dose of MSM coated on HAp for osteoinduction at in vitro level. We now have included this information in Materials and Methods.

Comment #3

You have tested sulphur concentration by using ICP-OES. So, does it mean the release of sulphur ion or MSM?

Response to Comment #3

Thank you for kind your comment. We have used the ICP-OES method to detect Sulphur ion in accordance to the previous literature as Sulphur would be an indicator for MSM. Still, we agree that actual test is detection of Sulphur and not MSM. Hence, we have corrected subtitles in Materials and Methods 2.5.

Comment #4

Fig 1 - different colours represent which colours?

Response to Comment #4

Thank you for kind comment. In Figure 1, the element mapping of color meant that green, orange, and purple are in order calcium, phosphorus, and sulfur. We revised that the color of component indicated the Figure 1.

Reviewer 3 Report

This work is well researched and well written. However, I still have two questions:

(1) In the introduction, the author mentioned that " However, there have been limited reports of MSM application in bone tissue engineering".  Please state the novelty of current work in a more clear way.

(2) Did the mechanical properties of porous scaffold also show great potential?

Thanks.

Author Response

Review #3

This work is well researched and well written. However, I still have two questions:

Comment #1

In the introduction, the author mentioned that " However, there have been limited reports of MSM application in bone tissue engineering".  Please state the novelty of current work in a more clear way.

Response to Comment #1

Thank you for kind your comments. According to advice your comment, we revised the sentence in the line 49 to 52. The sentence was revised as below:

Before changing the sentence:

" However, there have been limited reports of MSM application in bone tissue engineering"

After changing the sentence:
“Although MSM was well known in the osteogenic differentiation, the application of MSM has been few reports in the biomedical field such as bone tissue engineering and regenerative medicine.

Comment #2

Did the mechanical properties of porous scaffold also show great potential?

Response to Comment #2

Thank you for your valuable comments. The HAp scaffold fabricated by sponge replica method was like cancellous bone. According to previous studies, it was reported the adequate mechanical properties by such HAp scaffold. Still, the aim of study was the application of MSM in the HAp scaffold for osteogenic properties considering cancellous bone. We have included some of these details in Discussion as limitation and future studies.